# Metacoupling of Water Transfer: The Interaction of Ecological Environment in the Middle Route of China’s South-North Project

**DOI:** 10.3390/ijerph191710555

**Published:** 2022-08-24

**Authors:** Qingmu Su, Hsueh-Sheng Chang, Xiang Chen, Jingjing Xiao

**Affiliations:** 1School of Architecture and Planning, Fujian University of Technology, Fuzhou 350118, China; 2Department of Urban Planning, National Cheng Kung University, Tainan 70101, Taiwan; 3Institute of Urban-Rural Planning & Design, Xiamen University, Xiamen 361001, China; 4Department of Geology, National Taiwan University, Taibei 106216, Taiwan

**Keywords:** water transfer, metacoupling, ecological environment, Middle Route of China’s South–North Water Transfer Project (SNWTP-MR), sustainable development

## Abstract

At present, nearly half of the population of China live in water-deficient areas where water needs to be transferred from surrounding or remote water sources to meet local water demands. Although the water transfer project has alleviated the demands for water in the water-deficient areas, and brought water-supply income to water source regions, it has also posed some cross-regional negative impacts, including the changes in the original ecology within the water source, the impacts on the downstream water demands, and the risk of biological invasion in the distant water receiving areas. Therefore, it can be seen that the impact of water transfer is complicated and will be manifested in various aspects. The Middle Route of China’s South–North Water Transfer Project (SNWTP-MR), as the world’s largest cross-watershed water transfer project, exerts particularly important effects on regional sustainable development; however, it also produces complex interactions within the ecological environment itself, downstream and in the distant water receiving cities. Thus, this work attempts to apply a metacoupling analysis framework of water transfer to explore the ecological interaction of water transfer in SNWTP-MR on each system. The metacoupling framework can be divided into intracoupling, pericoupling and telecoupling. This study focuses on the analysis of the causes and effects of the intracoupling, pericoupling and telecoupling of SNWTP-MR from the perspective of ecological values and ecological risks. We found that the coupling of water transfer brings about 23 billion yuan of ecological service value to the water source annually, but also increases the internal ecological risk index by 9.31%, through the calculation of changes in land use; secondly, the power generation benefit significantly increases, and the flood control standards have shifted from once-in-20 years to once-in-a-century. However, the ecological risks are also significant, such as poor water quality, eutrophication of water resources, competition for water between industry and agriculture, deterioration of waterway shipping, and threats to biodiversity, etc. Considering only water supply, the population carrying capacity of the water resource in distant water receiving cities is increased by 16.42 million people, which enhances the value of water resources and creates a cross-regional green ecological landscape belt. Nevertheless, the biological invasion and water pollution have greatly affected the safety of water supply. It can be seen that the cross-regional water transfer does not always damage the interests of the sending system and the spillover system while benefiting the receiving system; its impacts are complex and variable. Through this paper, it is hoped to provide a reference for the analysis of the ecological compensation, resource development and allocation in SNWTP-MR by revealing the metacoupling relationship of SNWTP-MR. This paper will provide new ideas for researching the metacoupling relationship, thereby offering valuable reference for the study of the interaction generated by large-scale water transfer.

## 1. Introduction

As sustainable development is a major challenge around the world, how to achieve the sustainability of urban water supply systems is one of the global concerns [1,2]. The world population reached 7.7 billion in May 2019 and is estimated to increase to 11.2 billion in 2100 [3]. However, the rapid urbanization and population growth will lead to a series of severe challenges, especially in terms of urban water security. The water sources in large cities only account for 1% of the world’s total water sources, but the urban population consumes 41% of the world’s water resources [4,5]. Therefore, large cities or places where the population gathers need a large amount of water transferred from surrounding areas or remote distant areas to meet the local water demands. China’s water resources are estimated to total 2800 billion m^3^, ranking fourth in the world. However, affected by natural factors and human activities, the water resources are unevenly distributed in different regions. For example, the water resources in the northern areas are unable to support the sustained and sound development of the economy, resulting in increasingly significant predicaments. To solve this problem, the Chinese government has implemented regional water transfer projects through dam construction, water diversion, etc., among which the most influential is China’s South-North Water Transfer Project (SNWTP). However, there is no systematic answer as to whether other impacts will be caused if the water is transferred; considering the large scale of the project, it is necessary to have a comprehensive analytical framework to evaluate the impact of the water transfer project on river ecology and urban ecology [6,7].

China’s South-North Water Transfer Project (SNWTP) is by far the world’s largest cross-watershed water transfer project, and the Middle Route of China’s South–North Water Transfer (SNWTP-MR) can alleviate the problem of insufficient water in the north to a large extent, especially in Beijing, Tianjin, Hebei, and Henan where high water risks exist [8]. However, under multiple influences such as the unique natural regional unit of the water source (at the junction of three provinces and the north-south boundary of China), the human activities (regional cultivation and reservoir expansion) and policy implementation (water withdrawal and immigration in the middle route), SNWTP-MR has led to complex environmental changes and variations in hydrological processes. Moreover, the resulting evolution and migration of the ecological environment system directly affects the security of the ecological environment and socio-economic sustainable development. At the same time, stakeholders often lack cross-sectoral coordination [9]. Therefore, it is necessary to study the interaction caused by SNWTP-MR on water source areas, downstream and in distant water receiving cities to reveal the systematic coupling and sustainable development of the ecological environment.

Many scholars have explored these issues from different aspects. There are studies focusing on the interactions within the coupling system, or concentrating on the social and economic interactions between coupling systems [10,11]. Su et al. (2021) divided the metacoupling of the water transfer project into intracoupling, pericoupling and telecoupling, opened the internal structure of the traditional data envelopment analysis (DEA) system, and analyzed the efficiency of water transfer [12]. Chang et. al. (2021) used inter-regional land exchange as the basis for fair distribution, and analyzed the cross-regional impact of stormwater flow through local coupling and telecoupling [13]. In general, the conceptual framework of telecoupling provides a much-needed comprehensive method for systematic research which clearly examines the interaction of the coupling of human and natural systems on time and space scales; it is an effective framework to solve the sustainability of urban water resources, and is also an important reference framework for this paper [6,14].The framework consists of five interrelated components: the coupling of human and natural system (system); the material, information and energy flow within the system (flow); agent promoting the flow (agent); driving causes of flow (cause); and the effects of the flow (effects). The direction of the flow determines that the system can be considered as a sending system (for example, the sending system in this study mainly refers to the watershed range of the Danjiangkou Reservoir), the receiving system (for example, the distant cities that receive water from the sending system and the downstream of the Danjiangkou Reservoir), and the spillover system (for example, the counties and cities along the SNWTP-MR) [15]. This study abandons the “single dimension” mode in the traditional research on coupling, and combines with the conceptual framework of telecoupling to build the metacoupling framework of water transfer in SNWTP-MR, including the intracoupling within the system, and the pericoupling and telecoupling between two systems.

This study attempts to answer the following questions under the framework of exploring the metacoupling of water transfer from SNWTP-MR: How can we systematically analyze the interaction among the water source area, downstream areas, and the distant water receiving cities in SNWTP-MR? How can the ecological value and uncertain risks be studied based on technologies such as GIS? How should we further quantitatively reveal the coupling relationship among various systems? In this study, these problems and related issues are investigated under the new metacoupling framework to attempt to illustrate the ecological coupling of SNWTP-MR with limited information.

## 2. Materials and Analytical Framework

### 2.1. Study Area

China’s SNWTP-MR is 1430 km long and provides domestic water to 155 billion m^2^ of land, including Beijing, Tianjin, Hebei and Henan (Figure 1A). Since the route was officially put into use in December 2014, as of 12 December 2018, the total amount of water transfer was 22.2 billion m^3^ [16]. The water source is the Danjiangkou Reservoir on the Han River (the tributary of the Yangtze River) and its upstream area, including 95,200 km^2^ of the drainage divides in 43 counties in Henan, Hubei and Shaanxi Province (Figure 1B). The downstream watershed of the Han River covers an area of 43,800 m^2^ (Figure 1C), including 30 counties in Henan and Hubei provinces.

### 2.2. Data Sources

First, remote sensing map data were obtained from the United States Geological Survey (https://earthexplorer.usgs.gov/, accessed on 1 October 2019), and 10 image data of 30 m × 30 m of Landsat in 2013 and 2019 were downloaded, respectively; they were used for evaluating the vulnerability of the ecological environment of the SNWTP-MR water source area. SNWTP-MR was officially put into use in December 2014. Secondly, the information collected from the SNWTP-MR website, Beijing Statistical Yearbook, Hebei Statistical Yearbook, Henan Statistical Yearbook, Danjiangkou Reservoir data, etc., was used as the main source of data in this study.

### 2.3. Analytical Framework

The metacoupling system is a set of two or more coupled systems that interact internally as well as nearby and far away, facilitated by agents affected by various causes with various effects [17]. The entire metacoupling system consists of a sending system, receiving system, and spillover system (Figure 2), among which the sending system provides materials and information for the entire coupling; the receiving system obtains materials and information from the sending system and provides energy and capital for the sending system; while the spillover system is the area affected by the transfer of materials and information, and is the medium for the transfer between the receiving system and the sending system. Further, the metacoupling framework can be divided into three parts: intracoupling, pericoupling and telecoupling [17] (Figure 2). 

Intracoupling refers to the internal mutual influence of the sending system. For example, the sending system of SNWTP-MR is Danjiangkou Reservoir, and intracoupling studies the relationship of ecological changes in the water source area’s own system.

Pericoupling refers to the mutual influence between the sending system and the receiving system, emphasizing the influence of the sending system on the surroundings, and there is no need for conduction between them through other media. For example, the pericoupling of SNWTP-MR refers to ecological coupling in the same watershed.

Telecoupling is similar to pericoupling, but it emphasizes the cross-regional relationships. The sending system and the receiving system need to pass through the spillover system to have a relationship. For example, the Danjiangkou Reservoir of SNWTP-MR needs to be connected to the northern water-receiving city through the construction of a canal.

Due to the complex impact of the SNWTP-MR project, this article uses a system perspective to decompose the impact, which will help to present the problem and simplify the internal complexity of SNWTP-MR. We segmented metacoupling to facilitate the understanding of the interactions among them, so as to assess the sustainable development of the ecological environment of SNWTP as a whole. However, the metacoupling framework can integrate the impacts that may not be quantifiable or measurable, and expand the existing methods. For quantifiable impacts, the framework can provide the quantitative results of the coupling process, and for non-quantifiable impacts, it can improve the qualitative understanding of the relationship between agent and flow [18].

The metacoupling framework (Figure 3) begins with the analysis of the ecological environment in intracoupling, pericoupling, and telecoupling, including the main influences, the causes and ways of coupling, to clarify the ecological values and ecological risks brought by these potential factors. Intracoupling evaluates the vulnerability of the ecological environment in the water source area of SNWTP-MR according to the data in typical years selected, analyzes the temporal and spatial pattern and evolution trend of the ecological environment, and pre-processes the data on Plowland, Vegetation, Water, Artificial surface, and Bare Soil of the water source area with the help of remote sensing and geographic information system (GIS), so as to reveal the spatial coupling of Danjiangkou waters and the ecological effects of the construction of SNWTP-MR. These effects include impact on the ecological value (the economic value of land type) and ecological risk (improvement of ecological risk index), which are then measured by economic evaluation of ecosystem services and intensity coefficients of ecological risk theory. The Danjiangkou Reservoir is the boundary to divide the sending system (upstream water source area) and the receiving system (downstream area). Pericoupling reveals that the ecological value and ecological risk of the two systems are not always mutually matching and benignly complementary under the coupling of environmental ecological effects. For example, a situation may occur where migration may mitigate the carrying capacity of ecological environment in upstream, but poses a burden to the environmental carrying capacity downstream. We explored the ecological values from the aspects of power generation benefits, seasonal water transfer, water quality, flood control benefits, and the amount of water abandonment of floods. We also investigated the ecological risks from issues such as eutrophication of water, the deterioration of shipping capacity, the competition for water between industry and agriculture, and biological threats. A clear understanding of the interaction between the upstream and downstream aspects of the Danjiangkou Reservoir is beneficial to dealing with the contradictory relationship between them; telecoupling divides the sending system, the spillover system and the receiving system. The remote water transfer will affect the safety of water supply, population carrying capacity, water quality and groundwater level in water receiving areas, and produce ecological risks such as water source pollution and biological intrusion. The spillover system is directly and indirectly affected by water pollution and agricultural irrigation, and it also exerts pressure on the water quality protection and water development of the sending system. By analyzing the telecoupling relationship, we can understand the interaction among them in terms of ecological value and ecological risk. The framework encompasses with the environmental interaction of metacoupling, and based on the interaction of ecological environments in the coupling relationship, we can reveal how to promote the sustainable development under this interaction of coupling.

## 3. Application of the Metacoupling Framework in SNWTP-MR

### 3.1. Metacoupling Framework Analysis

Under the framework of metacoupling, we redefine the multi-system coupling relationship of SNWTP-MR as follows, focusing mainly on system, agent, flow, cause and effect.

#### 3.1.1. System

Intracoupling is mainly the interaction among different types of land use in Danjiangkou water source, emphasizing the mutual transformation among Plowland, Vegetation, Water, Artificial Surface and Bare Soil. Since pericoupling is in the cross-boundary watershed, the different demands of upstream and downstream will affect the impact of water transfer in the watershed. Therefore, we use the sending system and the receiving system to analyze the pericoupling relationship between upstream and downstream caused by SNWTP-MR. In telecoupling, the watershed range of Danjiangkou Reservoir is the sending system, the distant water receiving city (Province) (Henan, Hebei, Tianjin, Beijing) is the receiving system, and the counties and cities along the route constitute the spillover system.

#### 3.1.2. Agent

The metacoupling of SNWTP-MR mainly involves the participation of local and central government, enterprises and farmers in the construction of the reservoir, the transfer of water, the protection of the environment and the coordination among the various systems. To help overcome the project’s technological and engineering challenges, thousands of experts—representing government agencies, corporations, banks, and non-governmental organizations—have been recruited both domestically and internationally, in addition to millions of laborers. At the central government level, there is the SNWTP Construction Committee Office under the State Council. At the local and regional levels, there are construction committees in each affected city and province [15].

#### 3.1.3. Flow

The flow of metacoupling involved in water transfer is mainly the volume of water transfer, for example, the multi-year average volume of water transfer is 9.7 billion m^3^ per year, of which the volume of water transfer to Henan, Hebei, Beijing, and Tianjin is 3.71, 3.47, 1.24, and 1.02 billion m^3^, respectively, and the necessary volume of runoff in downstream areas [19]. Secondly, there is a spatial flow of industrial and agricultural pollutants with the water transfer. In addition, the water delivery at the pump station also leads to the flow of energy. Thirdly, there is a spatial transfer of population in the areas affected by the project construction. Finally, the construction and operation of the SNWTP-MR also results in the flow of funds. The main flows of intracoupling include water, money, and population; the main flows of pericoupling are water, money, and energy; and the main flows of telecoupling are water, money, pollution, and energy.

#### 3.1.4. Cause

The reason for the intracoupling of the whole SNWTP-MR is mainly attributed to the change in the nature of the land which was caused by the construction of Danjiangkou Reservoir, and the development and utilization of land due to the spatial transfer of population and local economic development. In addition, land development will also be restricted due to the strengthened government focuson ecological protection. From the perspective of the dam itself, the reason for pericoupling is that because the dam body of the Danjiangkou Reservoir is raised, a large part of the water storage is used for the downstream water supply, so that the water for use in the downstream area is reduced. From the perspective of the water receiving area and downstream area, the reason for pericoupling is that the receiving water area and the downstream area have different purposes for the use of water resources, and there is a competitive relationship between the two. From the perspective of upstream and downstream, the interests of upstream and downstream users are different, resulting in conflicts and contradictions in water allocation. Under these multiple influences, the services and the willingness for development in the watershed will be inevitably affected. The potential reasons for telecoupling include four aspects: ecology, economy, politics and technology. From the perspective of ecology, there are excessive exploitation of groundwater and serious water pollution in the north, so the spatial water transfer is conducive to improving the carrying capacity of ecological environmental resources. From the point of view of the economy, there is a large distribution of agriculture and industry in the north, accompanied by a lack of water resources, so in order to achieve the sustainable development of the economy, it is a feasible option to seek alternative water resources. From a political perspective, the distribution of water resources and population in the north and the south is uneven; the remote water transfer in SNWTP-MR can increase taxes for local government and provide employment opportunities. From a technical point of view, the terrain along the route is high in the south and low in the north, which means that natural runoff can be achieved through the influence of gravity in most parts of the route, and there are not many places where a pumping station is needed for water delivery [20]; thus, water transfer is feasible from the aspects of geographical elevation and energy. 

#### 3.1.5. Effects

Intracoupling affects the hydrogeomorphic characteristics, causing the pressure of population transfer, and limiting the development of industry and agriculture to a certain extent. Obviously, the result of coupling will also promote ecological conservation (see Table 1 below for details). The impact of pericoupling is that the expansion of the Danjiangkou Reservoir brings both opportunities and challenges to the ecological and economic use of water resources, and also influences the transformation of regional industry and society (see Table 2 below for details). The impact of telecoupling is manifested in multiple aspects, including economy, society, ecology, etc. Water transfer can put pressure on the finances of different systems or increase their income, and also will threaten or promote the environmental carrying capacity of each system. At the same time, water transfer causes a threat to biological survival and the pressure on migration (see Table 3 below for details). In this paper, we focus on the ecological value and ecological risk generated by SNWTP-MR. Therefore, a description of detailed quantitative and qualitative effects is conducted below from the perspective of ecological value and ecological risk, in order to better present the impact of metacoupling.

### 3.2. From the Perspective of Ecological Value

The intracoupling of SNWTP-MR refers to the interactions between different types of land use in Danjiangkou water source areas, so it is necessary to analyze their patterns of spatial and temporal change pattern. Based on the satellite remote sensing images of the Danjiangkou water source in 2013 and 2019, we classified the land use type by the Threshold method of eCognition 8.7 software (Developed by Trmible, Sunnyvale, CA, USA) and the Pretreatment method of ENVI 5.1 software (Developed by Exelis Visual Information Solutio, Boulder, CO, USA) (including radiation calibration, atmospheric correction, image mosaic, image cropping). According to the theories of Costanza et al. and Yang Guoqing [21,22], when calculating the ecological service value and ecological risk, land use is mainly divided into five categories, and this study also continues this classification. The categories include: Plowland (paddy field, dry land), Vegetation (woodland, bush, open woodland, other woodland, high coverage grass, medium coverage grass, low coverage grass), Water (canals, lake, reservoir pond), Artificial Surface (urban land, rural settlement), and Bare Soil (sandy ground, Gobi saline-alkali land, bare earth, bare rock texture) (Figure 4 and Figure 5). The “accuracy assessment” analysis index was used for accuracy verification. The specific method is to compare and test the drawings analyzed in this study using the vector files of the identified land types on the software. The verification result is: the accuracy of the classification in 2013 is about 87%, about 85% in 2019, so the accuracy is high. This can be seen from the changes in land use (Figure 6). From the red circles A and B in Figure 6, it is also found that there is obvious land development; at the same time, the expansion of the reservoir increases the water area significantly (the red circle C in Figure 6), which will provide a solid foundation for remote water transfer.

Table 3 and Table 4 show the changes in land use. We use the spatial dynamic degree of change in land use to reflect the speed of land use change under coupling. The spatial dynamic degree refers to the ratio of the product of the translation of this type into the sum of other types and the translation of other types into the sum of this type to the number of certain land types at the end of the period, which is used to reflect the rate of change [23]. It can be seen from Table 5 that the water transport by SNWTP-MR leads to an increase in the water level of the Danjiangkou Reservoir, forcing the population migrated due to the Reservoir to develop more of the highland. This can also be indirectly reflected by the increase in the area of the Plowland and the Artificial Surfaces. At the same time, the area of Vegetation has not changed much in the past six years, which also demonstrates that the protection of the water source of Danjiangkou Reservoir has exerted certain effects. The changes in intracoupling will also affect the ecological value and ecological risk in the area.

Based on the above-mentioned changes in land use, the ecological service value of the water source of SNWTP-MR can be measured. In terms of the measurement of the ecological value of intracoupling, Costanza proposed the theoretical framework of economic evaluation of ecosystem services, which provides a perspective for the measurement of the relative ecological value of land use types [21,24]. Costanza et al. classified land use into 17 types, and the ecological services of each type include four major categories, namely, supply service, regulation service, support service and cultural service, and nine sub-services. Specifically, the relative value of each unit of arable land is linked to the market price of its output value, and the value of other types of land depends on its relative importance to arable land (equivalence coefficient) [25]. Built-up land is considered incapable of providing ecosystem services [26,27]. For example, the output value per hectare in 2013 was 25,837 yuan, while in 1995 it was 13,350 yuan, so the relative increase was 1.94 times. We can calculate the ecological service value of the year according to the corresponding ratio. According to the theory proposed by Costanza et al. and the average grain price per hectare for the current year, Plowland, Vegetation, Water, and Bare Soil of 1 hm^2^ can provide the ecological value of 4.7279, 1.8774, 10.7772, and 3.303 million yuan, respectively. Therefore, the increase in the water area due to the expansion of the Danjiangkou Reservoir can create an ecological service value of 1.307 billion yuan; on the whole (Table 6), the coupling of the internal land use types has increased the value of 206.86 billion yuan provided in 2013 to the ecological services value of 229.937 billion yuan, an increase of 23 billion yuan. Therefore, although the construction of SNWTP-MR has changed various land types to some extent, as a whole, it is still conducive to providing ecological value in water sources. (The ecological value emphasized in this article generally refers to the services that ecology can provide, including quantifiable and non-quantifiable values, such as the value of ecological services. Ecological service value refers to the ecological value that can be measured by money. Benefits refer to effects and incomes, which can be economic, social and environmental benefits, and include the expansion of ecological value, such as flood control benefits.)

In terms of pericoupling, first it is found that the power generation benefit of water resources and seasonal water transfer have been improved. For example, the total length of the Danjiangkou dam is 2.5 km, the maximum dam height of the project is 97 m, the installed capacity is 900,000 kilowatts, and the multi-year average annual power generation is 4 billion kilowatt hours, which can bring an income of 2.292 billion yuan to the water source of Danjiangkou Reservoir every year if calculated at the lowest electricity price of 0.573 yuan in the downstream Wuhan city. During the main flood season from July to September each year, the water level can be regulated to below the flood control level, and seasonal water transfer can be realized. At the end of the flood season in October each year, the required electricity can be generated and water can be stored; at the same time, the stored water can also be used to maintain seasonal normal runoff [28,29]. Second, the amount of water loss (abandonment) due to the flood is reduced. Suppose we subtract the downstream water consumption from the water output of the reservoir and divide the result by the downstream water consumption as the water abandonment rate. The maximum water consumption of the downstream cities of 8.17 billion m^3^ is selected as the water consumption in the lower reaches of the Han River [30]; the water output of Danjiangkou Reservoir in 2013 before the water transfer was 26.71 billion m^3^. After the water transfer in 2014, the water output was 15.45 billion m^3^, so the water abandonment rate was reduced by 137.8%, and the utilization of water resources has been significantly promoted. Third, flood control benefits: After the Danjiangkou Reservoir was raised in height, the reserved flood control capacity of the Reservoir was increased by 3.28 billion m^3^ (in summer flood season) and 2.51 billion m^3^ (in autumn flood season). Meanwhile, by combining the total flood diversion capacity of the downstream flood diversion areas, the flood control standard in the lower reaches of the Han River can be changed from once-in-20-years to once-in-a-century (controlling the once-in-a-century flood as in 1935) to eliminate the flood threats to more than 700,000 people downstream [31].

In terms of telecoupling: First, in this study, it is found that water supply can create more economic value. The water transfer increases the income of the sending system, and at the same time guarantees the water use of the receiving system and alleviates the shortage of water. From Table 7, we can see that SNWTP-MR can bring the water supply value of 9.85 billion yuan to the water source every year, which will make up for the cost on engineering development and the loss of economic value caused by water source protection. Second, the system creates an ecological corridor. The water in Danjiang River passes through sluice gates, canals and ecological forests on both sides, building a green corridor with a length of more than 1430 km and a width of several tens of meters to form a cross-regional green ecological landscape belt. The spillover system can use the corridor to bring excess water into the corridor for use in water-deficient areas, so as to achieve the balance of water transfer in a small area. The creation of the ecological corridor will enhance the greening level and ecological benefits in various systems. Third, (3) the water transfer improves the population carrying capacity of water resources. According to the research results of the Sustainable Development Strategy Research Group of China’s 21st-century Agenda Management Center, with the coefficient of water supply capacity adopted as 0.39 (technical parameter), the calculation results (Table 7) show that the water transfer from SNWTP-MR will increase the population carrying capacity of Henan, Hebei, Beijing and Tianjin by 5.99, 5.57, 2.67 and 2.18 million people, respectively, in 2019. Fourth, the telecoupling promotes the benefits of flood control, humidity increase, temperature reduction and environmental purification. According to the relevant research, the value of the absorption of sulfur dioxide and nitrogen oxides and of dust retention by forest is determined to be 215.6, 6.0, 21.7 kg/hm^2^, respectively. The sewage charges are 1.2, 0.6, 0.2 yuan/kg, respectively, and the width of the SNWTP-MR greenway is 50 m [32,33]. It can be seen from Table 7 that the benefit of environmental purification in Henan, Hebei, Beijing, and Tianjin is increased to 97, 79, 10, and 3 million yuan, respectively. Fifth, the water quality of urban living water and rivers and lakes is improved. The water inflow from SNWTP-MR is blended with the water in northern counties and cities, which reduces the hardness of the water. The hardness of the tap water in Beijing is reduced from 380 mg∙L^−1^ to 120–130 mg∙L^−1^ [34]. At the same time, excess water can be used to supplement the water volume of rivers and lakes. For example, Hebei Province uses the water from SNWTP-MR to replenish 0.07 billion m^3^ of water to the Hutuo River and Qili River. Sixth, there is an increase the agricultural water and the area of waters. When providing both domestic and industrial water, SNWTP-MR also takes into account the water for ecological environment and agriculture along the route; at the same time, SNWTP-MR can replenish water in rivers and lakes and water sources in areas along the route, which is conducive to increasing the area of waters. For example, SNWTP-MR has supplemented an accumulation of 171 million m^3^ of water to Beijing Miyun Reservoir, effectively inhibiting the decline of water in Miyun Reservoir [35]. Finally, the system alleviates the problem of over-exploitation of groundwater. The distant water receiving cities (receiving system) effectively alleviate the deterioration of groundwater ecology by replacing the local groundwater source with water brought by SNWTP-MR. The groundwater level in the water-supplement areas has increased by different degrees. For example, in 2017, the average buried depth of groundwater in Beijing increased by 0.53 m year-on-year, and the groundwater level in Tianjin increased by 0.17 m [34].

Summary. The benefits brought by water transfer in SNWTP-MR are reflected in multiple aspects (Figure 7). The expansion of the Danjiangkou Reservoir in the water source changed the original land type, and the relative ecological value of the land use type has improved. At the same time, the expansion and water transfer of the Danjiangkou Reservoir broke the original spatial and temporal configuration pattern of water resources, which brings some additional ecological values to the upstream and downstream areas, including the increase in total power generation caused by the increase in reservoir water storage, the enhanced ability of seasonal water transfer, and the effective coordination of flood control in the region. These, in turn, promote the reduction of the water abandonment rate of floods in the sending system (water source), so that the power generation benefit can be improved. SNWTP-MR focuses on solving the problem of water shortage in the north.The transfer through the spillover system can provide economic value of water supply, improve the water quality of urban domestic water and rivers and lakes, alleviate the problem of over-exploitation of groundwater, improve the population carrying capacity of water resources and promote the benefits of flood control, carbon fixation and oxygen release, and air purification. These further bring economic value of water supply to the water source, raise the awareness of ecological conservation, and also create ecological corridor for the spillover system and alleviate the shortage of agricultural irrigation water. The analysis of the ecological value brought by SNWTP-MR through the metacoupling framework is helpful for us to understand the coordinated development among various systems.

### 3.3. From the Perspective of Ecological Risk

Based on the above-mentioned changes in land use type, the changes in the ecological risk of the SNWTP-MR water source can be measured. In terms of intracoupling, the ecological risk caused by the intensity of the changes in different land types is cumulative. Therefore, it is necessary to judge the accumulated risks from the perspective of the overall system and establish an empirical connection between land use structure and ecological risk. The researches in this aspect are mainly conducted through establishing the intensity coefficient of ecological risk to transform the land use types into ecological risk variables [22]. As can be seen from Table 8, the ecological risk index of the Danjiangkou Reservoir water source was 0.165 in 2013, and 0.18 in 2019, an increase of 9.31%. Therefore, the construction of SNWTP-MR has increased the level of ecological risk in the water source, and the ecological vulnerability has been destroyed to some extent. At the same time, the population transfer and resettlement from the submerged area of the reservoir (300,000 people) has caused a large amount of ecological land upstream of the water source to be used for the construction of new towns, which will lead to further ecological damage, and the disturbance of human activities will exacerbate ecological fragility.

In terms of pericoupling:First, the reduction of the downstream water flow causes ecological problems. From 2012 to 2016, the outflow water of Danjiangkou Reservoir was 36.27, 26.71, 15.45, 28.19, and 15.01 billion m^3^, respectively, showing a decreasing trend, which caused the water level in the lower reaches of the Han River to drop by between 0.3 m and 1 m [40]. The reduction of water volume also resulted in the eutrophication of water resources and the gradual deterioration of water quality in the downstream of the Han River. The reduction of downstream water flow has given rise to the worse overall shipping capacity. After the water transfer, the average annual navigable time is only 121 days, and the navigation guarantee rate is significantly reduced to 33% [41]. At the same time, the decreased water level will result in the deterioration of downstream water quality. As the amount of inflow is reduced, the flow rate will slow down, which makes the self-purification capacity of the water body in the lower reaches of the Han River drop drastically. At the same time, industry and agriculture pump a large amount of water, and a large part of this water directly enters the river, further polluting the river, while the urgent need for the improvement of water quality in the downstream will also put more pressure on the water quality protection in the upstream water source. In addition, the shortage of water intensifies the competition for water between industry and agriculture, and the rate of average water supply guarantee of each water plant has dropped by 34.7% [40]. For the downstream watershed, the reduction of water volume and the increase of the dam body have led to the destruction of the ecological balance of aquatic organisms in the Han River, and threat to biodiversity. Second, biodiversity and fish stocks are threatened. After the water transfer, the number of fish spawning grounds has been reduced by 25, the spawning time is delayed, and the total amount of fish is reduced by 1/4 or more, among which the production of wild fish is reduced by 50%, greatly destroying fish diversity and output value. Besides, the variety of hygrophyte in the lower reaches of the Han River will be affected to a certain extent, and the area of swamp vegetation, such as Aceh Weed Swamp, Valerian Swamp, Reed Marsh, and Calamus Marsh will decrease [42,43]. 

In terms of telecoupling: First, the primary impact is the biological invasion, which threatens species survival. The construction of water transfer canals has created channels for biological exchanges. Some fish in the Danjiangkou Reservoir may enter other reservoirs and watershed areas along the SNWTP-MR, which will cause changes in the number of fish species in other reservoirs and watersheds. Secondly, the water resources along the route are polluted, threatening the safety of water supply. For distant water receiving cities, SNWTP-MR adopts open channels for water delivery under direct sunlight, with long routes, shallow water and low water flow rate. The water temperature changes with air temperature (from 20 °C in September to May to above 30 °C in July and August), which will be conducive to algae reproduction and lead to a further increase in algal density and the risk of algae odor [35]. There are clams in the Danjiangkou Reservoir, with a short reproductive cycle, and the larvae belong to plankton, while the adults grow on the hard substrate, attached by the foot. They often block the water pipeline and the water plant filter, threatening the safety of water use. Meanwhile, clams are tolerant to the temperature in the north, so there is a risk that they will migrate northward along the SNWTP-MR [44,45]. 

Summary From the perspective of ecological risk (Figure 8), the expansion of the Danjiangkou Reservoir in the water source has changed the original land type, which increases the ecological risk index and further enhances the ecological vulnerability. At the same time, the water transfer to the north will inevitably lead to a significant reduction in downstream water flow, which will then result in poorer water quality, eutrophication of water resources, competition between industry and agriculture for water, deterioration of conditions for waterway shipping, threats to the diversity of aquatic organisms and fish survival, further intensifying the ecological conflicts between upstream and downstream. Moreover, the most direct ecological impact of remote water transfer on the distant water receiving city is the problem of biological invasion and water pollution, which in turn forces the water source to strengthen the monitoring of biological invasion. The conflict in water use between the downstream area and the distant water receiving city also aggravates the conflicts in the coordination of water resources in the water source. By clarifying the ecological risk of the water transfer in SNWTP-MR via a metacoupling framework, we can take necessary measures to prevent and mitigate its negative effects.

## 4. Discussion—The Impact of the Metacoupling on the Sustainability of SNWTP-MR

SNWTP-MR brings risks to the ecological environment while creating benefits for the ecological environment. It is difficult to say who should take the responsibility and who can get the benefits. The metacoupling framework can help us to better understand and manage the sustainability of regional water transfer, identify gaps between systems and hidden problems based on the direct and indirect positive and negative feedback, as illustrated in Figure 9. The coupling of water transfer does not always damage the benefits of the sending system and the spillover system while benefitting the receiving system, but presents a more complex and varied situation. It is known from the previous sections that the intracoupling can bring about 23 billion yuan of ecological service value. The pericoupling significantly reduces the flood control pressure in the downstream, and the telecoupling can compensate the water volume for the spillover system, create a green corridor and bring growth of domestic water, population carrying capacity and economy to the receiving system. The metacoupling of SNWTP-MR can fully present the interrelationship between systems. When we clarify the value and risks it brings and try to overcome the risks, a water transfer project such as SNWTP-MR can bring win-win benefits to each system.

The systematic illustration of the metacoupling of the SNWTP-MR provides a framework for the reasonable allocation of water resources and promotes the sustainability of water transfer. Long-distance water transfer, which reduces downstream water inflow, coupled with different interests of upstream, downstream and the distant water receiving city, affects the willingness of all parties to protect rivers and pay for the ecological services, and intensifies the conflicts among all the parties in the competition for water [6]. For example, the distant water receiving cities consumed 9.7 billion m^3^ of water in the Danjiangkou Reservoir, almost one-third of the water volume in the Danjiangkou water source, which causes the changes in the land use type of the water source to produce a chain effect, and imposes tremendous pressure on the ecosystem services [46,47]. The annual decline in the downstream water volume has threatened the agricultural land and food production, and resulted in the deterioration of river channels to some extent.The role of the feedback and negative feedback of the metacoupling relationship provides useful information for long-distance water transfer, and offers a good opportunity to correct water allocation in annual water supply, so that the water allocation can achieve a positive balance to support social and economic development while protecting and improving the future environment. Applying the metacoupling framework to understand the sustainability issues can yield valuable insights to identify both beneficiaries and losers; thus, the overexploitation and unreasonable allocation of resources can be avoided [16].

The analysis of the benefits and disadvantages to various systems in SNWTP-MR, brought by the metacoupling framework, provides a theoretical reference for who should take greater responsibility for ecological compensation. We know that SNWTP involves not only resettlement but also ecological and economic costs. Migration resettlement and food demand may lead to reclamation of agricultural land, degradation of forests and surrounding grasslands, which has been verified in our intracoupling framework. The reduction in downstream water volume also causes real pressure; although the water crisis can be alleviated in water receiving city, the increase in water use will also reduce the efficiency of water consumption. In this aspect, the metacoupling framework can help us to estimate and determine who will compensate, what to pay and how much to pay [25]. From the perspective of fairness and justice, ecological compensation can offset the social and economic benefits given up by water delivering areas. The analysis of the causes and impacts of coupling can help us to further understand the social and equity issues related to water use and coordinate the conflict between future land use and ecological protection, so as to promote the sustainability of water transfer [18]. This study is helpful for diagnosing which links have gone wrong in the sustainability of water transfer, which needs to be adjusted, and what linkage effects will be produced after the adjustment. Therefore, this study is conducive to the sustainability of long-term water transfer.

The analysis results of the metacoupling of SNWTP-MR helps to clarify the problems and challenges faced by different organizations, systems and agents, and can help them to take separate or joint actions. We know that SNWTP-MR has changed the economic structure and the method for ecological treatment in water sources, and also promoted the transformation of regional industry and society. The complexity of water transfer has transcended the water trading and investment itself. Therefore, the metacoupling framework is conducive to uncovering the complex interactions between water resources and environmental sustainability. In advancing the sustainable development of SNWTP-MR, joint actions can be taken to reduce costs, improve water quality, reduce pressure on water use and environment, promote social technological innovation and the implementation of the payments for watershed service plans, and enhance the complementarity of water resources [48].

Previous studies have mainly discussed SNWTP-MR ecological payment, water transfer policies, water environment governance, and water market incentive coordination studies. Few studies have been carried out on the metacoupling impact of the entire water transfer from a global perspective [49,50,51]. Therefore, the innovation of this research lies in solving the problem of the interaction of ecological environment of the pericoupling and telecoupling of water transfer and sustainable development under the new integrated framework (metacoupling), providing a new research idea for coupling relationships. At the same time, it reveals the metacoupling relationship of SNWTP-MR and the mutual influence of the ecological environment, which can provide a reference for SNWTP-MR, and also provide a valuable reference for other large-scale water transfer interactions. For example, Australia is considering sending large amounts of water over long distances from the north to the south of the country [52]. The cross-regional coupling relationship is more complex than the local coupling and requires a lot of resource integration. Metacoupling can find many research gaps and can conduct a comprehensive and in-depth analysis [17,53]. Therefore, the metacoupling framework can also be used for transnational and trans-regional research on ecosystem services, food trade, information dissemination, energy and species invasion.

This research still faces some research limitations. First, we are aware of the fact that it is challenging to study the interaction among multiple systems at the same time. Although some conclusions have been drawn by systematically analyzing the metacupling of SNWTP-MR with limited data, there is still more work to do to clarify the intricate relationship in reality. The biggest challenge is that the data are not easy to be established and acquired, which means that many environmental and socioeconomic impacts are not yet quantitatively measured. In the future, more efforts need to be put into predicting the value and risk brought by the metacoupling relationship of SNWTP-MR. Second, the Metacoupling framework is still a conceptual framework. The unified evaluation of the integration of intracoupling, pericoupling and telecoupling is still a problem. Subsequent research can introduce some quantitative methods to conduct a unified evaluation of the metacoupling systems, such as data envelopment analysis or multi-region input and output models. Third, Differences in the time period taken by the remote sensing map will lead to a certain deviation between the changes in water volume and other data and the real measurement data. Therefore, this study has a certain error range, to which readers should pay attention.

## 5. Conclusions

This paper uses the metacoupling analysis framework of water transfer to reveal the influence of the interaction of ecological environment of SNWTP-MR. Thus, the following conclusions can be drawn:(1)By analyzing intracoupling, it can be shnown that water transfer has caused spatial and temporal changes in the land use of the Danjiangkou Reservoir, which affects the ecological value that it provides and exacerbates the internal ecological risk. Considering only land changes, the construction of SNWTP-MR has created an ecological service value of 23 billion yuan for the water source of Danjiangkou Reservoir, but also increased the original ecological risk index by 9.31%, and the ecological vulnerability has been changed to some extent.(2)By analyzing pericoupling, it can be found that the upstream and downstream have different purposes for water use, mainly because the expansion of the reservoir produces different ecological value and ecological risk for the upstream and downstream. The expansion of Danjiangkou Reservoir has increased the total power generation of the Reservoir, increasing the revenue of generation by 2.292 billion yuan; at the same time, the ability for seasonal water transfer has also been greatly enhanced, so that the downstream flood control standards have been improved to once-in-100-years, and the threat of flooding has been greatly reduced. However, the ecological risks are also present in many aspects, such as poor water quality, eutrophication of water resources, water competition between industry and agriculture, deterioration of waterway shipping conditions, and threats to biodiversity.(3)By analyzing telecoupling, the interaction among the sending system, the receiving system and the spillover system can be seen. From the perspective of ecological value, the sending system obtains a large amount of income from water transfer; the receiving system has greatly improved the urban domestic water consumption. For example, the water volume for consumption has been increased by 9.5 billion m^3^, and the hardness of tap water has also dropped from 380 mg∙L^−1^ to 120–130 mg∙L^−1^, and the problem of over-exploitation of groundwater has been alleviated. The population carrying capacity of the water resources has increased by 16.42 million people, and the spillover system has created a cross-regional green ecological landscape belt that can create a value of 189 million yuan of environmental purification benefits. From the perspective of ecological risk, both biological invasion and water pollution have become problems that all systems need to face together.

The metacoupling framework systematically discusses the values and risks brought by various systems of the SNWTP-MR, and provides a theoretical reference for who should take more responsibility for the ecological compensation. This may help avoid excessive development and unreasonable allocation of resources for this purpose.

## Figures and Tables

**Figure 1 ijerph-19-10555-f001:**
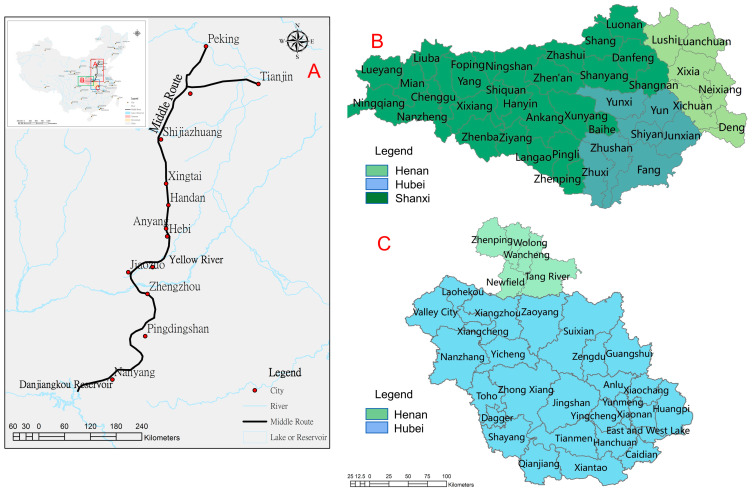
Range of influence of SNWTP-MR. ((**A**) is the whole process of water transfer in the SNWTP-MR; (**B**) is the scope of the water source area for water transfer; (**C**) is the downstream of the water source area).

**Figure 2 ijerph-19-10555-f002:**
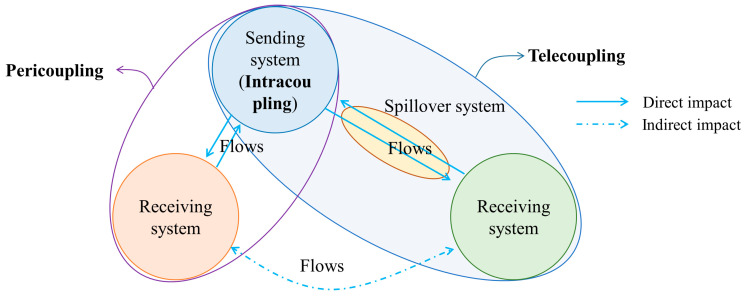
Metacoupling relationship.

**Figure 3 ijerph-19-10555-f003:**
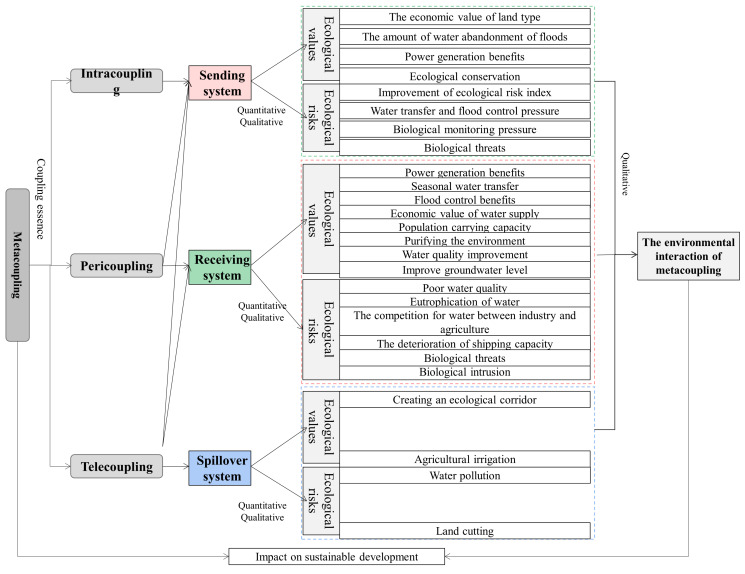
Research framework of SNWTP-MR metacoupling.

**Figure 4 ijerph-19-10555-f004:**
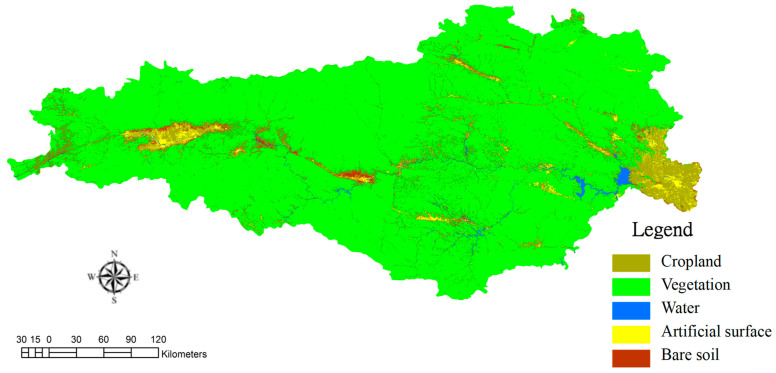
Land use of Danjiangkou water source in 2013.

**Figure 5 ijerph-19-10555-f005:**
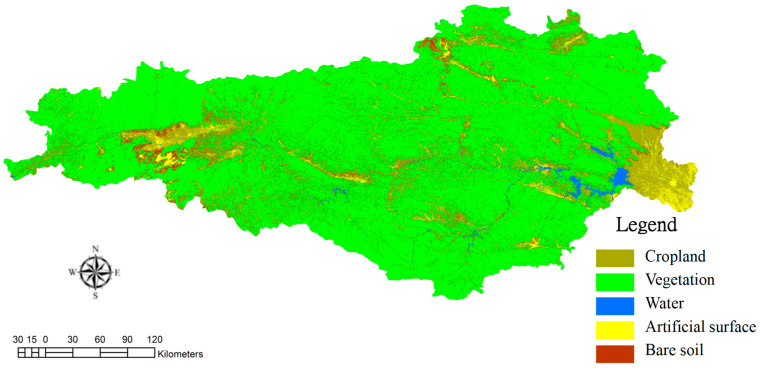
Land use of Danjiangkou water source in 2019.

**Figure 6 ijerph-19-10555-f006:**
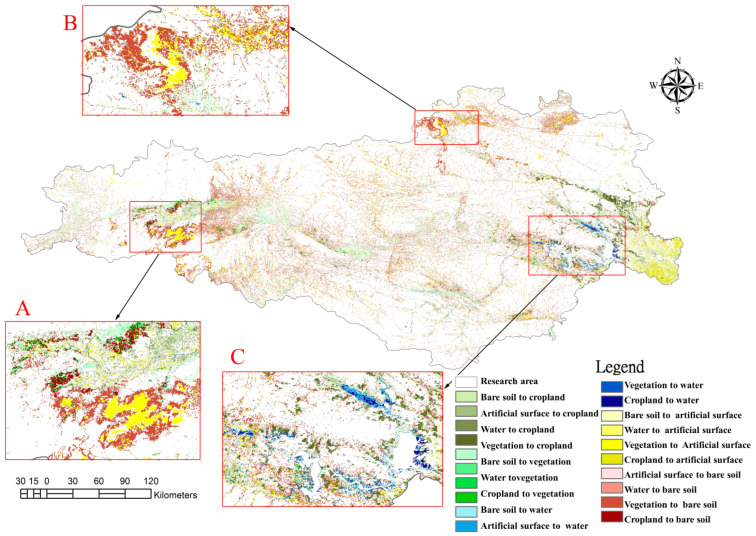
Total changes in land use from 2013 to 2019. (The circle (**C**) represents the increased area of the reservoir. Circle (**A**) and circle (**B**) indicate artificial surfaces added in 2019 over 2013).

**Figure 7 ijerph-19-10555-f007:**
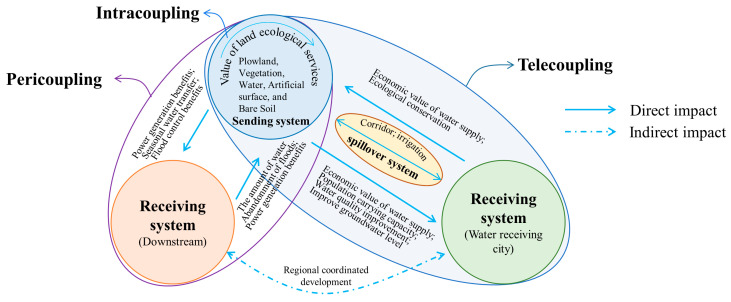
Schematic diagram of SNWTP-MR metacoupling ecological value.

**Figure 8 ijerph-19-10555-f008:**
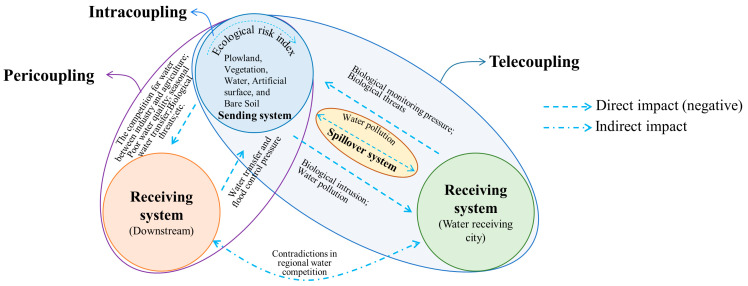
Ecological risk of the metacoupling of SNWTP-MR.

**Figure 9 ijerph-19-10555-f009:**
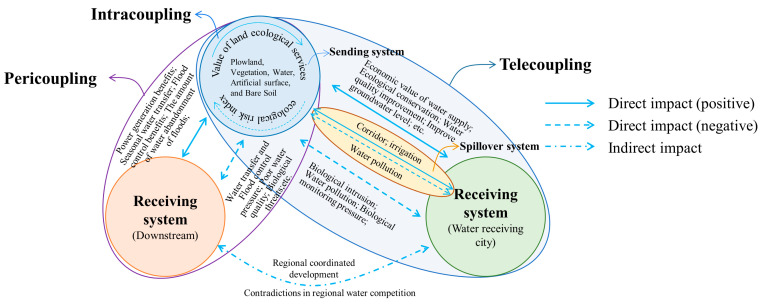
The ecological benefit feedback of the metacoupling of SNWTP-MR.

**Table 1 ijerph-19-10555-t001:** Analysis of the intracoupling framework of SNWTP.

	Sending System
System description	The watershed range of Danjiangkou Reservoir
Causes	Construction of Danjiangkou Reservoir, spatial transfer of population, local economic development, government’s interest in ecological protection
Agents	Central and local government, enterprises, farmers
Flows	Water, money and population
Effects (−)	The reservoir region immigration and resettlement cause ecological damage (relocation of 300,000 people). The hydrogeomorphic features have been changed, which increases the ecological risk. Industrial, agricultural and fishery development is restricted, thus farmers’ incomes are reduced.
Effects (+)	The water storage in the reservoir is increased, which improves the utilization of water resources. Local awareness of ecological protection has been aroused, and the ecological destruction rate in the water source is effectively contained.

**Table 2 ijerph-19-10555-t002:** Pericoupling of SNWTP.

	Sending System	Receiving System
System description	The watershed range of Danjiangkou Reservoir	Counties and cities in the downstream of Han River
Causes	For the SNWTP-MR, the dam body of the Danjiangkou Reservoir is raised, which reduces downstream water for use. The water receiving areas have different purposes for water consumption from the downstream areas, and there is a competitive relationship between the two. The interests of upstream and downstream users are different.	
Agents	Central and local government, enterprises, farmers	
Flows	Water, money and energy	
Effects (−)	The pressure on flood control is increased; the transfer and resettlement of the population from the reservoir-inundated region brings huge burden; the hydrological and geomorphological characteristics have been changed.	Water flow is reduced; water quality is decreased; flow rate is slowed, shipping capacity is reduced; biobalance is destroyed, and biodiversity is threatened; water consumption in industry and agriculture is affected.
Effects (+)	The water storage in reservoir is increased; total power generation is improved; the value of tourism development is increased; the utilization efficiency of water resource is increased.	The regional electricity guarantee is increased; the flood control pressure is reduced; the seasonal water allocation can reduce the threat to water supply caused by extreme weather; industries with high demands of water can carry out industrial transformation and upgrading.

**Table 3 ijerph-19-10555-t003:** Telecoupling of SNWTP.

	Sending System	Receiving System	Spillover System
System description	The watershed range of Danjiangkou Reservoir	Henan, Hebei, Tianjin, Beijing	Counties and cities along the route
Causes	Geographical elevation allows water transfer, the terrain along the route is high in the south and low in the north, and naturalrunoff can be realized in most parts of the route; There is an uneven distribution of water resources and population in the north and south. There is a large distribution of agriculture and industry in the north, accompanied by the lack of water resources; There is excessive groundwater exploitation and serious water pollution in the north.		Spatial transfer of water
Agents	Central and local government, enterprises, farmers		
Flows	Water, money, pollution, energy		
Effects (−)	Migration out of the reservoir region causes environmental burden and brain drain; threats to species’ survival.	Biological invasion; elevated water, nitrogen and phosphorus nutrients; water use risk; human and financial input; payment for water resources.	Water quality decline; the need for water resources protection along the route increases financial pressure.
Effects (+)	Income from water transfer; dam power generation; ecological conservation; tourism development. A large amount of water can be transferred out during the flood season to reduce the threat of floods.	Agricultural irrigation water; domestic water; reduced groundwater exploitation; increased population capacity and economic growth; water quality improvement;	Ecological corridor; tourism development; use of corridor to achieve water transfer balance in small range.

**Table 4 ijerph-19-10555-t004:** Matrix of the land change and transfer in the water source of Danjiangkou Reservoir in 2013–2019.

2013–2019	Plowland (km^2^)	Vegetation (km^2^)	Water (km^2^)	Artificial Surface (km^2^)	Bare Soil (km^2^)	Total	$\mathrm{Reduction}\;({\mathrm{km}}^{2})\;(\Delta U_{\mathrm{re}})$
Plowland (km^2^)	2305.81	142.07	84.96	886.73	183.38	3602.95	1297.14
Vegetation (km^2^)	1161.04	88,234.62	253.31	2565.63	7672.90	99,887.50	11,652.89
Water (km^2^)	40.66	186.96	640.01	109.25	96.69	1073.58	433.57
Building (km^2^)	562.75	286.15	125.23	1063.31	237.32	2274.75	1211.44
Bare Soil (km^2^)	308.12	1773.17	91.38	953.67	1362.08	4488.42	3126.34
Total	4378.38	90,622.96	1194.89	5578.60	9552.37		
$\mathrm{Increments}\;({\mathrm{km}}^{2})(\Delta U_{\mathrm{in}})$	2072.57	2388.34	554.88	4515.29	8190.29		
Change (km^2^)	775.43	−9264.54	121.31	3303.85	5063.95		

**Table 5 ijerph-19-10555-t005:** Spatial dynamic degree of land change in the water source of Danjiangkou Reservoir in 2013–2019.

Name	Formula	Plowland	Vegetation	Water	Buildings	Bare Soil
spatial dynamic degree	$C=\frac{\Delta U_{\mathrm{in}}\times \Delta U_{\mathrm{re}}}{U}$ *C* : Rate of change over a period of time; $\Delta U_{\mathrm{in}}$: Other types translate to the sum of this type; $\Delta U_{\mathrm{re}}$: This type translates to the sum of other types; *U*: The number of certain land types at the end of the period	0.77	0.15	0.83	1.03	1.18

**Table 6 ijerph-19-10555-t006:** Ecological service value of intracoupling.

Name	Formula	Plowland (km^2^)	Vegetation (km^2^)	Water (km^2^)	Artificial Surface (km^2^)	Bare Soil (km^2^)	Total		Sources
Value of ecological services	$V_{S}=\sum_{i=1}^{m}T_{i}\times P_{i}$ $T_{i}$: Area of type i land type; *P_i_*: Ecological value of each land type	207.01	1701.36	128.78	0.00	31.55	2068.69	2013 year (100 million yuan)	[22]
		382.83	1610.93	224.99	0.00	80.62	2299.37	2019 year (100 million yuan)	

Note: The units of money used in this study are expressed in Renminbi (RMB).

**Table 7 ijerph-19-10555-t007:** Benefits created by SNWTP-MR for water receiving provinces and cities.

	Index Province (City)	Henan	Hebei	Beijing	Tianjin	Total	Sources
Data	Multi-year average (billion m^3^)	37.7	34.7	12.4	10.2	95	[19]
	Water price (yuan/m^3^)	0.37	0.97	2.33	2.16		[36]
	Per capita water consumption (m^3^/person)	245	243	181	182		[37]
	Pipe length (km)	731	596	80	25	1432	
Benefit	Formula						
Benefits of water supply	VQ=P×Q (billion m^3^) Q: Total water transfer	13.95	33.66	28.89	22.03	98.53	[38]
Benefits of population carrying capacity of water resources	$W=\frac{\alpha \times Q}{C}$ (Ten thousand people) α: Coefficient of water supply capacity C:The amount of water used per capita	599	557	267	218	1642	[39]
Benefit environmental purification	$Ve=L\times W\times \sum_{i=1}^{n=3}\left(Qi\times Pi\right)$ (Billion yuan) *Q_i_*: Respectively indicate sulfur dioxide, nitrogen oxides and dust retention P_i_: Sewage charge price W: Width of SNWTP-MR greenway	0.97	0.79	0.1	0.03	1.89	[32,33]

**Table 8 ijerph-19-10555-t008:** The ecological risk index of intracoupling.

Name	Formula	Plowland (km^2^)	Vegetation (km^2^)	Water (km^2^)	Artificial Surface (km^2^)	Bare Soil (km^2^)	Total	Year
ecological risk index	$E=\sum_{i=1}^{m}\frac{T_{i}\times \delta_{i}}{T}$ $\delta_{i}$: Intensity coefficient of ecological risk of the i-th land type; *T*: area of Total land; *T_i_*: Area of type i land type	0.010	0.135	0.002	0.014	0.004	0.165	2013
		0.012	0.122	0.002	0.035	0.009	0.180	2019

## Data Availability

All data generated or analyzed during this study are included in this article.

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
