# Peer review of "Metacoupling of Water Transfer: The Interaction of Ecological Environment in the Middle Route of China’s South-North Project"

_ijerph, 2022, doi:10.3390/ijerph191710555_

Round 1

Reviewer 1 Report

The journal's scope includes this topic. Errors are evident in the current draft, however. To support the recommendation for publishing, appropriate modifications must be made.

  • Increase the DPI of the image in Fig 1, make it  at least 600 or greater,  in Fig1, the section  B, and C are not readable.

  • Although the introduction section is well-connected, a few relevant and latest publications may be referred to in the introduction section demonstrating intracoupling, telecoupling and pericoupling systematic approaches employed for water distribution around the world. 

  • How Direct negative impacts accessed? Discuss about migration, LULC change, biological thereats in little detail

  • What is rationale behind selecting five categories: Plowland, Vegetation, Water, Artificial surface, and Bare Soil? Dont you think the large share comes under vegetation which is vague.  The forest, grass, different forest types is missing? Not much has been talked about forests and forest ecosystem services.

  • In Fig. 4 and 5 enlarge the caption stating the name of the region 

  • A DEM of the study region could have shown the topographical features, the study heavely rely upon LULC only, discuss why? 

  • The valuation of ecological services is made by Costanza method, Elaborate a little more about the approach and also discuss how inflation in monetary term over time is taken care of?

  • Line number 117 and 120 the area of Han River watershed is in square meter check unit 

  • Line number 628 [stream flood control standards have been improved to once-in-20-years, and the threat of flooding has been greatly reduced.] check the sentence at another place it has been said once in 100 years.

  • Table 7, see the superscript of the units, see this for the whole manuscript.

  • Publication discussion sections should summarize the most important findings, explain how those results fit into a broader body of work in science and discuss any issues or differences between this work and other published works that have been studied.

  • Make the conclusion section crisp, concise and connected 

Author Response

Dear Reviewer,

We have greatly appreciated the reviewers’ efforts. Comments and feedback were very constructive and able to improve the quality of the manuscript.

My Manuscript: ijerph-1865956

Article titled:  "Metacoupling of Water Transfer: the Interaction of Ecological Environment in the Middle Route of China’s South - North Project"

Modified parts of the manuscript have been shown in red font. The following is a summary of the reviewers' comments.

Sincerely,

Qingmu Su

Reviewer: 1

Suggestions:1. Increase the DPI of the image in Fig 1, make it  at least 600 or greater,  in Fig1, the section  B, and C are not readable.

Author reply: Thank you for your suggestion. We have increased the resolution of the pictures.

Suggestions:2. Although the introduction section is well-connected, a few relevant and latest publications may be referred to in the introduction section demonstrating intracoupling, telecoupling and pericoupling systematic approaches employed for water distribution around the world.

Author reply: Thanks to the reviewer's reminder, we have added some current literature on coupling research to improve the rationality of this research framework and coupling. For example line 92-97.

“Su et. al. (2021) divided the metacoupling of the water transfer project into intracoupling, pericoupling and telecoupling, opened the internal structure of the traditional DEA system, and analyzed the efficiency of water transfer [12]. Chang et. al. (2021) used inter-regional land exchange as the basis for fair distribution, and analyzed the cross-regional impact of stormwater flow through local coupling and telecoupling [13].”

Suggestions:3. How Direct negative impacts accessed? Discuss about migration, LULC change, biological thereats in little detail

Author reply: Thanks to the reviewer for your suggestions, we have added the negative effects of migration, LULC changes, and biological phenomena. Such as line 480-488, 490-494, 517-523.

line 480-488 “In terms of intracoupling, the ecological risk caused by the intensity of the changes in different land types is cumulative. Therefore, it is necessary to judge the accumulated risks from the perspective of the overall system and establish an empirical connection between land use structure and ecological risk. The researches in this aspect are mainly conducted through establishing the intensity coefficient of ecological risk to transform the land use types into ecological risk variables [22]. As can be seen from Table 8, the ecological risk index of the Danjiangkou Reservoir water source was 0.165 in 2013, and 0.18 in 2019, an increase of 9.31%.”

line 490-494 “At the same time, the population transfer and resettlement in the submerged area of ​​the reservoir (300,000 people) has caused a large amount of ecological land upstream of the water source to be used for the construction of new towns, which will lead to further ecological damage, and the disturbance of human activities will exacerbate ecological fragility.”

line 517-523 "Biodiversity and fish stocks are threatened. After the water transfer, the number of fish spawning grounds has been reduced by 25, the spawning time is delayed, and the total amount of fish is reduced by 1 / 4 or more, among which the production of wild fish is reduced by 50%, greatly destroying fish diversity and output value; besides, the variety of hygrophyte in the lower reaches of the Han River will be affected to a certain extent, and the area of swamp vegetation, such as Aceh Weed Swamp, Valerian Swamp, Reed Marsh, and Calamus Marsh will decrease[42, 43].”

Suggestions:4. What is rationale behind selecting five categories: Plowland, Vegetation, Water, Artificial surface, and Bare Soil? Dont you think the large share comes under vegetation which is vague.  The forest, grass, different forest types is missing?

Author reply: Thanks to the reviewer for your suggestions, we divide them into these five categories based on the research on ecological service value and ecological risk by Costanza et al. and Yang Guoqing below. And we add in parentheses the land-use types included in each category. Specifically, such as line 315-321.

“According to the theories of Costanza et al. and Yang Guoqing [21, 22], when calculating the ecological service value and ecological risk, land use is mainly divided into five categories, and this study also continues this classification. Such as: Plowland (Paddy field, Dry land), Vegetation (Woodland,Bush, Open woodland, Other woodland, High coverage grass, Medium coverage gras, Low coverage grass), Water (Canals, Lake, Reservoir pond), Artificial surface (Urban land, Rural settlement), and Bare Soil (Sandy ground, Gobi Saline-alkali land, Bare earth, Bare rock texture) (Figure 4 and 5).”

Suggestions:5. In Fig. 4 and 5 enlarge the caption stating the name of the region

Author reply: Thanks to the reviewer's suggestion, we have revised the name of the title. For example "Figure 4 Land use of Danjiangkou water source in 2013 and Figure 5 Land use of Danjiangkou water source in 2019"

Suggestions:6. The valuation of ecological services is made by Costanza method, Elaborate a little more about the approach and also discuss how inflation in monetary term over time is taken care of?

Author reply: Thanks to the reviewer for the reminder that this was an oversight in our writing, and we added the way we converted it at the time. Such as line 361-368.

“Specifically, the relative value of each unit of arable land is linked to the market price of its output value, and the value of other types of land depends on its relative importance to arable land (equivalence coefficient) [25]. But built- up land is considered incapable of providing ecosystem services [26, 27]. According to the theory proposed by Costanza et al. and the average grain price per hectare for the current year, Plowland, Vegetation, Water, and Bare Soil of 1hm2 can provide the ecological value of 4.7279, 1.8774, 10.7772, and 3.303 million yuan, respectively. ”

Suggestions:7. Line number 117 and 120 the area of Han River watershed is in square meter check unit 

Author reply: Thanks to the reviewer for your suggestions. We modified it.

Suggestions:8. Line number 628 [stream flood control standards have been improved to once-in-20-years, and the threat of flooding has been greatly reduced.] check the sentence at another place it has been said once in 100 years.

Author reply: Thanks to the reviewer for your careful observation. It was an oversight in our writing and we changed it to 100 years.

Suggestions:9. Table 7, see the superscript of the units, see this for the whole manuscript.

Author reply: Thanks to the reviewer for your careful observation. We reviewed the entire article and revised it.

Suggestions:10. Publication discussion sections should summarize the most important findings, explain how those results fit into a broader body of work in science and discuss any issues or differences between this work and other published works that have been studied.

Author reply: Thanks to the reviewer for your suggestions. We add differences from previous studies, contributions and application prospects of this study. Such as line 628-645.

“Previous studies have mainly discussed SNWTP-MR ecological payment, water transfer policies, water environment governance, and water market incentive coordination studies. Few studies have been carried out on the metacoupling impact of the entire water transfer from a global perspective [49-51 ]. Therefore, The innovation of this research lies in solving the problem of the interaction of ecological environment of the pericoupling and telecoupling of water transfer and sustainable development under the new integrated framework (metacoupling), providing a new research idea for coupling relationship. At the same time, it reveals the metacoupling relationship of SNWTP-MR and the mutual influence of the ecological environment, which can provide a reference for SNWTP-MR, and also provide a valuable reference for other large-scale water transfer interactions. For example, Australia is considering sending large amounts of water over long distances from the north to the south of the country [52 ]. The cross-regional coupling relationship is more complex than the local coupling and requires a lot of resource integration. Metacoupling can find many research gaps and can conduct a comprehensive and in-depth analysis [17, 53]. Therefore, the Metacoupling framework can also be used for transnational and trans-regional research on ecosystem services, food trade, information dissemination, energy and species invasion.”

Reviewer 2 Report

1. Lines 199-202: The author claims that: “The framework ends with the environmental interaction of metacoupling, and based on the interaction of ecological environment of the coupling relationship, we can reveal how to promote the sustainable development under this interaction of coupling.” However, the article lacks an explanation of the sustainable development path by metacoupling.

 2. Lines 328-329: Table 4, ”Matrix of the land change and transfer in the water source of Danjiangkou Reservoir in 2013-2019.” The water area of Danjiangkou is 1022.75 square kilometers. The data in the table seems to be quite different from the actual situation, which will make the readers feel confused. Please clarify it reasonably.

 3. Line 353: Table 6 classifies the land use types into five categories: plowland, vegetation, water, artificial surface, and bare soil. However, what subtypes are included in each classification that needs to be presented to the readers, such as garden land, forest land, pasture land, and transportation land?

 4. Lines 339-341: “According to the theory proposed by Costanza et al., Plowland, Vegetation, Water, and Bare Soil of 1hm2 can provide the ecological value of 4.7279, 1.8774, 10.7772, 340 and 3.303 million yuan, respectively.” Is it reasonable and scientific to directly use the value parameters of foreign research to assess the ecosystem service value caused by the change in land use type due to the South-to-North Water Diversion Project in China? Furthermore, those parameters are still more than 20 years old. If this method is adopted, at least the parameters need to be adjusted, taking into account the exchange rate, comparable value, and other factors, to show the difference in ecosystem service value caused by the social and economic development difference in various countries and regions.

Author Response

Dear Reviewer,

We have greatly appreciated the reviewers’ efforts. Comments and feedback were very constructive and able to improve the quality of the manuscript.

My Manuscript: ijerph-1865956

Article titled:  "Metacoupling of Water Transfer: the Interaction of Ecological Environment in the Middle Route of China’s South - North Project"

Modified parts of the manuscript have been shown in red font. The following is a summary of the reviewers' comments.

Sincerely,

Qingmu Su

Reviewer: 2

Suggestions:1. Lines 199-202: The author claims that: “The framework ends with the environmental interaction of metacoupling, and based on the interaction of ecological environment of the coupling relationship, we can reveal how to promote the sustainable development under this interaction of coupling.” However, the article lacks an explanation of the sustainable development path by metacoupling.

Author reply: Thanks to the reviewer for your suggestions. We have added the impact of this study on sustainable development in the Discussion section. Such as line 597-615.

“The analysis of the benefits and disadvantages brought by metacoupling framework to various systems in SNWTP-MR provides a theoretical reference for who should take greater responsibility for ecological compensation. We know that SNWTP involves not only resettlement but also ecological and economic costs. Immigration resettlement and food demand may lead to reclamation of agricultural land , degradation of forests and surrounding grasslands, which has been verified in our intracoupling framework; the reduction in downstream water volume also causes real pressure; although the water crisis can be alleviated in water receiving city, the increase in water use will also reduce the efficiency of water consumption. In this aspect, the metacoupling framework can help us to estimate and determine who will compensate, what to pay and how much to pay [25]. From the perspective of fairness and justice, ecological compensation can offset the social and economic benefits given up by water deliveri ng areas. The analysis of the causes and impacts of coupling can help us to further understand the social and equity issues related to water use and coordinate the conflict between future land use and ecological protection, so as to promote the sustainability of water transfer [18 ]. This study is helpful for diagnosing which links have gone wrong in the sustainability of water transfer, which needs to be adjusted, and what linkage effects will be produced after the adjustment. Therefore, this study is conducive to the sustainability of long-term water transfer.”

Suggestions:2. Lines 328-329: Table 4, ”Matrix of the land change and transfer in the water source of Danjiangkou Reservoir in 2013-2019.” The water area of Danjiangkou is 1022.75 square kilometers. The data in the table seems to be quite different from the actual situation, which will make the readers feel confused. Please clarify it reasonably.

Author reply: This study is based on the analysis of the surface conditions of remote sensing maps. Because of the different time periods taken by the remote sensing map, data such as water volume will change. At present, the water area in 2019 in the table is 1194.89 square kilometers, which has a certain error with the actual measurement. We therefore added it to our study constraints. Such as line 658-661.

"Differences in the time period taken by the remote sensing map will lead to a certain deviation between the changes in water volume and other data and the real measurement data. Therefore, this study has a certain error range, which requires readers to pay attention. "

Suggestions:3. Line 353: Table 6 classifies the land use types into five categories: plowland, vegetation, water, artificial surface, and bare soil. However, what subtypes are included in each classification that needs to be presented to the readers, such as garden land, forest land, pasture land, and transportation land?

Author reply: Thanks to the reviewer for your suggestions, we divide them into these five categories based on the research on ecological service value and ecological risk by Costanza et al. and Yang Guoqing below. And we add in parentheses the land-use types included in each category. Specifically, such as line 315-321.

“According to the theories of Costanza et al. and Yang Guoqing [21, 22], when calculating the ecological service value and ecological risk, land use is mainly divided into five categories, and this study also continues this classification. Such as: Plowland (Paddy field, Dry land), Vegetation (Woodland,Bush, Open woodland, Other woodland, High coverage grass, Medium coverage gras, Low coverage grass), Water (Canals, Lake, Reservoir pond), Artificial surface (Urban land, Rural settlement), and Bare Soil (Sandy ground, Gobi Saline-alkali land, Bare earth, Bare rock texture) (Figure 4 and 5).”

Suggestions:4. Lines 339-341: “According to the theory proposed by Costanza et al., Plowland, Vegetation, Water, and Bare Soil of 1hm2 can provide the ecological value of 4.7279, 1.8774, 10.7772, 340 and 3.303 million yuan, respectively.” Is it reasonable and scientific to directly use the value parameters of foreign research to assess the ecosystem service value caused by the change in land use type due to the South-to-North Water Diversion Project in China? Furthermore, those parameters are still more than 20 years old. If this method is adopted, at least the parameters need to be adjusted, taking into account the exchange rate, comparable value, and other factors, to show the difference in ecosystem service value caused by the social and economic development difference in various countries and regions.

Author reply: Thanks to the reviewer for the reminder that this was an oversight in our writing, and we added the way we converted it at the time. Such as line 361-368.

“Specifically, the relative value of each unit of arable land is linked to the market price of its output value, and the value of other types of land depends on its relative importance to arable land (equivalence coefficient) [25]. But built- up land is considered incapable of providing ecosystem services [26, 27]. According to the theory proposed by Costanza et al. and the average grain price per hectare for the current year, Plowland, Vegetation, Water, and Bare Soil of 1hm2 can provide the ecological value of 4.7279, 1.8774, 10.7772, and 3.303 million yuan, respectively. ”

Reviewer 3 Report

This paper is a case study for analyzing the metacoupling of water transfer in China. However, some questions have not been well addressed. Thus, my suggestion is major revision at this stage.

1. Line 299: please give the the composition type of vegetation. 

2. Are the land use/cover data field-validated?

3. I found that authors used the benefits (e.g., Yuan) to reflect ecosystem value. However, benefits were usually subject to inflation and change over time. So did the author use a fixed benefit to calculate the ecological service value for different years?

4. The reference section was not well organized. Please refer to the requirements of this Journal. 

5. Some grammatical errors and irregular English writing can be found in the full text. I suggested that authors seek a native speaker to improve them. 

Author Response

Dear Reviewer,

We have greatly appreciated the reviewers’ efforts. Comments and feedback were very constructive and able to improve the quality of the manuscript.

My Manuscript: ijerph-1865956

Article titled:  "Metacoupling of Water Transfer: the Interaction of Ecological Environment in the Middle Route of China’s South - North Project"

Modified parts of the manuscript have been shown in red font. The following is a summary of the reviewers' comments.

Sincerely,

Qingmu Su

Reviewer: 3

Suggestions:1. Line 299: please give the the composition type of vegetation.

Author reply: Thanks to the reviewer for your suggestions, we divide them into these five categories based on the research on ecological service value and ecological risk by Costanza et al. and Yang Guoqing below. And we add in parentheses the land-use types included in each category. Specifically, such as line 315-321.

“According to the theories of Costanza et al. and Yang Guoqing [21, 22], when calculating the ecological service value and ecological risk, land use is mainly divided into five categories, and this study also continues this classification. Such as: Plowland (Paddy field, Dry land), Vegetation (Woodland,Bush, Open woodland, Other woodland, High coverage grass, Medium coverage gras, Low coverage grass), Water (Canals, Lake, Reservoir pond), Artificial surface (Urban land, Rural settlement), and Bare Soil (Sandy ground, Gobi Saline-alkali land, Bare earth, Bare rock texture) (Figure 4 and 5).”

Suggestions:2. Are the land use/cover data field-validated?

Author reply: We mainly process the land use type based on the remote sensing map, and apply the accuracy verification on eCognition 8.7 software and ENVI 5.1 software to verify. Such as line 321-325.

“The “accuracy assessment” analysis index was used for accuracy verification. The specific method is to compare and test the drawings analyzed in this study using the vector files of the identified land types on the software. The verification result is: the accuracy of the classification in 2013 is about 87% , about 85% in 2019, so the accuracy is high.”

Suggestions:3. I found that authors used the benefits (e.g., Yuan) to reflect ecosystem value. However, benefits were usually subject to inflation and change over time. So did the author use a fixed benefit to calculate the ecological service value for different years?

Author reply: Thanks to the reviewer for the reminder that this was an oversight in our writing, and we added the way we converted it at the time. Such as line 361-368.

“Specifically, the relative value of each unit of arable land is linked to the market price of its output value, and the value of other types of land depends on its relative importance to arable land (equivalence coefficient) [25]. But built- up land is considered incapable of providing ecosystem services [26, 27]. According to the theory proposed by Costanza et al. and the average grain price per hectare for the current year, Plowland, Vegetation, Water, and Bare Soil of 1hm2 can provide the ecological value of 4.7279, 1.8774, 10.7772, and 3.303 million yuan, respectively. ”

Suggestions:4. The reference section was not well organized. Please refer to the requirements of this Journal. 

Author reply: We rechecked the references section to make it compliant with the journal's requirements.

Suggestions:5. Some grammatical errors and irregular English writing can be found in the full text. I suggested that authors seek a native speaker to improve them.

Author reply: Thanks to the reviewer for the reminder. We have asked a specialized English language agency to grammatically check the full text, and we will conduct a new round of grammar checking after the article is accepted.

Round 2

Reviewer 1 Report

  • Refer to line numbers 116-119 [The water source is the Danjiangkou 116 Reservoir on the Han River (the tributary of the Yangtze River) and its upstream area, 117 including 95,200 m2 of the drainage divides in 43 counties in Henan, Hubei and Shaanxi 118 Province (Figure 1-b). The downstream watershed of the Han River covers an area of 119 43,800 m2 (Figure 1-c), including 30 counties in Henan and Hubei provinces.] My earlier question is still unanswered...

An another paper on South-to-North Water Transfer Project (Middle Route) of China shows (Han River is situated between the Daba Mountain and the sosoutherniedmont of the Qinling Mountains. It’s approximately 1,577 km long and drains about 159,000  square km ). You said upstream area (95,200 m2 = .095 square km). It may be that the unit is not correct, recheck

  • As its   evident, benefits are typically indexed to inflation and susceptible to alter over time, the answer to comment 6 is not crisp and clear, for general audience, a little more elaboration is needed.

Author Response

Dear reviewer,

We have greatly appreciated the reviewers’ efforts. Comments and feedback were very constructive and able to improve the quality of the manuscript.

My Manuscript: ijerph-1865956

Article titled:  "Metacoupling of Water Transfer: the Interaction of Ecological Environment in the Middle Route of China’s South - North Project"

Modified parts of the manuscript have been shown in red font. The following is a summary of the reviewers' comments.

Sincerely,

Qingmu Su

Reviewer 1

Question 1: Refer to line numbers 116-119 [The water source is the Danjiangkou 116 Reservoir on the Han River (the tributary of the Yangtze River) and its upstream area, 117 including 95,200 m2 of the drainage divides in 43 counties in Henan, Hubei and Shaanxi 118 Province (Figure 1-b). The downstream watershed of the Han River covers an area of 119 43,800 m2 (Figure 1-c), including 30 counties in Henan and Hubei provinces.] My earlier question is still unanswered...

Reply: Thanks to the reviewers for the reminder. We have enlarged the text in the figure to make it clearer.

Question 2: An another paper on South-to-North Water Transfer Project (Middle Route) of China shows (Han River is situated between the Daba Mountain and the sosoutherniedmont of the Qinling Mountains. It’s approximately 1,577 km long and drains about 159,000  square km ). You said upstream area (95,200 m2 = .095 square km). It may be that the unit is not correct, recheck

Reply: Thanks to the reviewers for their careful observation. This is a mistake in our writing. We changed 95,200 m2 to 95,200 km2. Such as Line 130-132.

“ its upstream area, including 95,200 km2 of the drainage divides in 43 counties in Henan, Hubei and Shaanxi Province (Figure 1-b).”

Question 3: As its   evident, benefits are typically indexed to inflation and susceptible to alter over time, the answer to comment 6 is not crisp and clear, for general audience, a little more elaboration is needed.

Reply: Thanks to the reviewers for the reminder. We give an example to make it more intuitive, such as line 361-367.

“Specifically, the relative value of each unit of arable land is linked to the market price of its output value, and the value of other types of land depends on its relative importance to arable land (equivalence coefficient) [25]. But built-up land is considered incapable of providing ecosystem services [26, 27]. For example, the output value per hectare in 2013 was 25,837 yuan, while in 1995 it was 13,350 yuan, so the relative increase was 1.94 times. We can calculate the ecological service value of the year according to the corresponding ratio.”

Reviewer 2 Report

The author has modified the manuscript reasonably.

Author Response

Reviewer 2

The author has modified the manuscript reasonably.

Reply: Thanks for your valuable comments. It is very helpful to improve the quality of the article. thanks.

Reviewer 3 Report

The MS has been revised at the current stage. I think it can be accepted.

Author Response

Reviewer 3

The MS has been revised at the current stage. I think it can be accepted.

Reply: Thanks for your valuable comments. It is very helpful to improve the quality of the article. thanks.

This manuscript is a resubmission of an earlier submission. The following is a list of the peer review reports and author responses from that submission.